# Analysis of the Fungal Community in Ziziphi Spinosae Semen through High-Throughput Sequencing

**DOI:** 10.3390/toxins10120494

**Published:** 2018-11-25

**Authors:** Mengyue Guo, Wenjun Jiang, Jiaoyang Luo, Meihua Yang, Xiaohui Pang

**Affiliations:** Key Lab of Chinese Medicine Resources Conservation, State Administration of Traditional Chinese Medicine of the People’s Republic of China, Institute of Medicinal Plant Development, Chinese Academy of Medical Sciences & Peking Union Medical College, Beijing 100193, China; guomy0908@hotmail.com (M.G.); wenjunjiang0927@gmail.com (W.J.); jyluo@implad.ac.cn (J.L.); mhyang@implad.ac.cn (M.Y.)

**Keywords:** Ziziphi Spinosae Semen, fungal contamination, amplicon sequencing, toxigenic fungi

## Abstract

Ziziphi Spinosae Semen (ZSS) has been widely used in traditional Chinese medicine system for decades. Under proper humidity and temperature, ZSS is easily contaminated by fungi and mycotoxins during harvest, storage, and transport, thereby posing a considerable threat to consumer health. In this study, we first used the Illumina MiSeq PE250 platform and targeted the internal transcribed spacer 2 sequences to investigate the presence of fungi in moldy and normal ZSS samples collected from five producing areas in China. Results showed that all 14 samples tested were contaminated by fungi. Ascomycota was the dominant fungus at the phylum level, accounting for 64.36–99.74% of the fungal reads. At the genus level, *Aspergillus*, *Candida*, and *Wallemia* were the most predominant genera, with the relative abundances of 13.52–87.87%, 0.42–64.56%, and 0.06–34.31%, respectively. Meanwhile, 70 fungal taxa were identified at the species level. Among these taxa, three potential mycotoxin-producing fungi, namely, *Aspergillus*
*flavus*, *A. fumigatus*, and *Penicillium citrinum* that account for 0.30–36.29%, 0.04–7.37%, and 0.01–0.80% of the fungal reads, respectively, were detected in all ZSS samples. Moreover, significant differences in fungal communities were observed in the moldy and normal ZSS samples. In conclusion, our results indicated that amplicon sequencing is feasible for the detection and analysis of the fungal community in the ZSS samples. This study used a new approach to survey the fungal contamination in herbal materials. This new approach can provide early warning for mycotoxin contamination in herbal materials, thereby ensuring drug efficacy and safety.

## 1. Introduction

Herbal medicines, which are commonly used to prevent, diagnose, and cure diseases, have played an important role in health care since ancient times [1]. However, many studies have described the occurrence of mycotoxins in medicinal plants and herbal medicines from various countries [2,3,4], thereby attracting considerable attention worldwide due to drug efficacy and safety. For instance, 58 (8.29%) and 17 (2.43%) of the 700 herbal medicine samples in South Korea are aflatoxin B_1_ (AFB_1_)- and total aflatoxin (AF)-positive, respectively, and the AFB_1_ (up to 73.27 mg/kg) and total aflatoxin contents (up to 108.42 mg/kg) in some samples exceeded the legal limits (10 mg/kg) [2]. Both AFs and ochratoxin A (OTA) are detected in all *Glycyrrhiza uralensis* samples (six moldy and nine normal samples) that were collected from different areas in China [5]. Moreover, the transfer rate of OTA and AFs were investigated in decoctions of herbal medicines. The transfer rate of OTA from the three herbal medicines, namely Trichosanthis Semen, Eucommiae Cortex, and Rubi Fructus, to decoctions is 12.72–61.33% [6]. Considerable transfer rates of AFB_1_ (4.37–26.37%), AFB_2_ (9.64–47.68%), and AFG_2_ (7.26–115.36%) are also observed from the five herbal medicines (i.e., Lilii Bulbus, Hordei Fructus Germinatus, Nelumbinis Semen, Polygalae Radix, and Bombyx Batryticatus) to decoctions [7]. Chronic toxicity is the most common form of mycotoxicoses and is caused by low-dose exposure over an extended period, leading to cancers and other irreversible effects [8,9]. The presence of mycotoxins in the decoctions poses a considerable direct threat to consumer safety, because the ingestion of even extremely small amounts of mycotoxins can lead to many diseases or even death in humans and animals [10]. Among the approximately 400 recognized mycotoxins [11], AFs, OTA, fumonisins (FBs), zearalenone (ZEA), deoxynivalenol (DON), and patulin are important contaminants on humans in terms of health perspective [12,13]. AFB_1_ has been classified as a group 1 carcinogen by the International Agency for Research on Cancer (IARC) [14] and is risk factors for human hepatocellular carcinoma [15]. Meanwhile, OTA has been classified in Group 2B as possibly carcinogenic to humans by IARC [14].

Some species from the *Aspergillus* and *Penicillium* genera are AFs and OTA producers [3,16]. The detection of toxigenic fungi in herbal medicines has received considerable attention. Singh et al. [17] found that *Aspergillus* is the most prevailing genus that infects the raw materials of six medicinal plants from the herbal markets of Varanasi, and 13 out of 32 isolates of *Aspergillus flavus* were AFB_1_-positive. Another study showed that 90% of the 30 medicinal plant samples from Pakistan are contaminated with molds. The frequently isolated fungi predominantly consist of *A*. *flavus*, *A*. *niger*, *A*. *parasiticus*, and *Penicillium* spp., and 31% of the 47 isolates tested are toxigenic [3]. Fungal contamination in herbal medicines occurs during all the processes, including growing, harvesting, cleaning, transporting, and storage [18,19,20,21]. Under favorable temperature and humidity conditions, toxigenic fungi belonging to *Aspergillus*, *Penicillium*, and *Fusarium* produce mycotoxins [13]. Among these fungi, 21 species from *Flavi*, *Ochraceorosei*, and *Nidulantes* in *Aspergillus* produce AFs, while various *Aspergillus* and *Penicillium* species produce OTA [22,23]. The occurrence of toxigenic fungi in herbal medicines has the potential to produce mycotoxins. Therefore, the simultaneous detection of fungi, especially toxigenic fungi, in herbal medicines is important for early warning for mycotoxin contamination. 

Ziziphi Spinosae Semen (ZSS, suanzaoren in Chinese), which is derived from the dried ripe seed of *Ziziphus jujuba* Mill. var. *spinosa* (Bunge) Hu ex H. F. Chou, is widely used to treat insomnia and palpitation in traditional Chinese medicine [24]. Pharmacological studies have demonstrated that ZSS regulates immune function, ameliorates learning and memory, and has a hypnotic-sedative effect [25,26,27]. Ziziphi Spinosae Semen is mainly produced in the Henan, Shanxi, Liaoning, and Hebei provinces of China and sold throughout the country. ZSS samples are prone to fungal and mycotoxin contamination if improperly harvested, processed, and stored. At present, regulation has been set for the maximum limits of AFB_1_ (5 μg/kg) and the sum of AFB_1_, AFB_2_, AFG_1_, and AFG_2_ in ZSS (10 μg/kg) in Chinese Pharmacopeia, respectively [24]. The accurate and rapid differentiation of fungi in ZSS is important to guarantee the safe use of ZSS. 

At present, the identification of fungi in herbal medicines is mainly based on fungi isolation and culture. The morphological and microscopic features of pure isolates are generally combined with DNA sequences for a comprehensive analysis [4,28,29]. However, fungi isolation and culture is a complicated and time-consuming procedure with the risk of missing some strains, thereby leading to an imprecise characterization of the fungal diversities. Therefore, an accurate and rapid method to detect fungi in herbal medicines is urgently needed. High-throughput sequencing (HTS) is a growth-independent method that can provide mass data of the composition of mixed microbial communities in low abundances, such as soil, sediment, and air filter samples. HTS has been widely used in fungal ecology studies [30,31,32], thereby providing a new prospect for the detection of fungal diversity in herbal medicines. 

Here, we first investigated the presence of fungi in ZSS by using the Illumina MiSeq PE250 platform and demonstrated the diversity of fungal contamination in the ZSS samples to provide early warning for mycotoxin contamination in ZSS.

## 2. Results

### 2.1. Analyses of the Diversity of Fungal Communities in the ZSS Samples 

All 14 ZSS samples were successfully amplified by PCR for the internal transcribed spacer 2 (ITS2) sequences. A total of 912,941 ITS2 sequences that were longer than 200 bp were produced after excluding the chimeric sequences. The optimized sequences were divided into 210 operational taxonomic units (OTUs) after cluster analysis. 

Appendix A shows the calculated Chao 1, Shannon, and Good’s coverage to measure the α-diversity. High Chao 1 value demonstrated a large variation of species in each sample. High Shannon value indicated high community diversity in each sample. Meanwhile, the results of Good’s coverage, which is an estimator of sampling completeness, indicated good overall sampling with levels of >99.8%. Rarefaction curve analysis showed that all samples were almost parallel to the x-axis, thereby indicating that the obtained reads were sufficient to represent the overall fungal diversity (Appendix A). The fungal community in the WM group also had higher Chao 1 estimate and Shannon index than those of the FM group (Figure 1a). The number of unique and common OTUs for the two groups are shown in a Venn diagram (Figure 1b). The results showed that the WM group possessed more unique OTUs than the FM group.

For β-diversity, principal coordinate analysis (PCoA) showed that samples were clustered according to the presence of mold. This result was consistent with the results of hierarchical clustering analysis (Figure 1c–d). Significant differences were reported between the FM and WM groups (analysis of similarity, ANOSIM, *R* = 0.524 and *p* = 0.002; Appendix A). By contrast, the samples from the five producing areas showed an insignificant difference (Appendix A). 

### 2.2. Fungal Community Composition in the ZSS Samples

Three fungal phyla, namely, Ascomycota, Basidiomycota, and Mucoromycota, were identified in all ZSS samples. Among the three phyla, Ascomycota was the dominant fungus, accounting for 64.36–99.74% of the fungal reads. Meanwhile, other fungal phyla were detected with extremely low relative abundances (0–0.33%, Figure 2a). 

Further taxonomical classification detected 61 genera, and the genera with high relative abundance are shown in Figure 2b. Among these genera, *Aspergillus*, *Candida*, and *Wallemia* were the most common, with the relative abundances of 13.52–87.87%, 0.42–64.56%, and 0.06–34.31%, respectively. The differences in the fungal community compositions between the FM and WM groups were significant. The relative abundances of the *Aspergillus* species in the FM group (42.66–87.87%) were higher than those of the *Aspergillus* species in the WM group (13.52 to 49.67%). Meanwhile, the relative abundances of the *Candida* species in the FM group (0.42–23.95%) were relatively lower than those of the *Candida* species in the WM group (3.13–64.56%). The linear discriminant analysis effect size (LEfSe) algorithm, which was used to identify the different relative abundances of the fungal taxa in the two groups, showed the same results (Figure 2c). Compared with the WM group, *Aspergillus* was the most dominant genus in the FM group. Meanwhile, the WM group possessed a much higher proportion of *Candida*, *Meyerozyma*, and *Botryosphaeria* species than those in the FM group. 

A total of 162 fungal taxa were identified in the ZSS samples. Among these fungal taxa, 70 can be identified at the species level, while the remaining 92 can be resolved at the genus level or higher. Among the 70 accurately differentiated species, potential mycotoxin-producing fungi, namely, *A*. *flavus*, *A*. *fumigatus*, and *Penicillium citrinum* that account for 0.30–36.29%, 0.04–7.37%, and 0.01–0.80% of the fungal reads, respectively, were detected in all ZSS samples (Figure 2d). A higher percentage of potential mycotoxin-producing fungi was also detected in the WM group (1.14–36.37%) than that in the FM group (0.61–5.03%).

## 3. Discussion

### 3.1. Prevalence of Fungal Contamination in the Commercial ZSS Samples

Fungi are common natural contaminants of herbal medicines due to their widespread distribution in the air. Fungal contamination in medicinal herbs has received considerable concern from the public. An investigation [33] on the occurrence of fungi in medicinal herbs and spices from India showed that 92% samples are contaminated by fungi, and 47% of the contaminated samples exceed the permissible limits set by the World Health Organization. Kong et al. [4] also found that 14 functional foods and 10 spices from Chinese markets are infected by molds. Zheng et al. [34] also showed the widespread fungal contamination in 15 medicinal herbs collected from China. Meanwhile, Su et al. [35] found that 48 samples of eight root herbs from Chinese markets are all contaminated by fungi. In all these studies, *Aspergillus* and *Penicillium* are identified as the most common contaminants. In the present study, the results indicated that *Aspergillus* was the most abundant genus in the ZSS samples, followed by *Candida*, accounting for 13.52–87.87% and 0.42–64.56% of the total reads, respectively. In contrast to previous studies, *Penicillium* species were detected at relatively low abundances (<1%) in all ZSS samples, except for the LY01 sample (7.77%). The relationship between fungal species and herbal materials cannot be fully explained owing to the complicated contamination reasons [34,36]. *Aspergillus* and *Penicillium* species were the principal storage fungi [36]. The substrate composition and storage conditions, such as moisture content, aeration, and temperature, might be the factors to influence the fungal community [34,36]. 

Meanwhile, *Aspergillus*, *Penicillium*, *Fusarium*, and *Alternaria* are the most common contaminants in food and herbal medicines, which contain the majority of mycotoxin producers [3,16]. Potential toxigenic fungi, such as *A*. *fumigatus*, *A*. *flavus*, *A*. *tubingensis*, and *A*. *aculeatus*, were previously isolated in different herbal medicines [3,34,37]. In the present study, three potential mycotoxin-producing fungi, namely, *A*. *flavus*, *A*. *fumigatus*, and *P*. *citrinum* that are potential producers of AFB_1_, OTA, and citrinin, respectively, were identified in the ZSS samples. The high relative abundances of the potential mycotoxin-producing fungi ranging from 0.61–36.37% in ZSS poses a considerable threat to public health. The relative abundances of the potential toxigenic fungi in normal samples were much higher than those in the moldy samples. Therefore, evaluating the safety of herbal medicines according to the presence or absence of molds is unadvisable. 

The differences in the fungal community structure between the FM and WM groups were significant. *Candida* species were dominant in the normal ZSS samples, whereas *Aspergillus* was the predominant genus in the presence of molds. Meanwhile, all samples were clustered according to the initial grouping in PCoA, thereby indicating the meaningfulness of grouping samples according to the presence of microscopic molds. However, the differences in the fungal community structures among the SX, HN, LN, HB, and SD groups were insignificant, thereby demonstrating that the presence or absence of molds affected the fungal community structure in ZSS samples to a greater extent than the producing areas.

### 3.2. DNA Marker Selection to Analyze the Fungal Community in ZSS 

An effective molecular marker is the precondition for the application of DNA barcoding technique. Schoch et al. [38] evaluated six candidate DNA regions, namely, ITS, *LSU*, *SSU*, *RPB1*, *RPB2*, and *MCM7*, for fungal identification and proposed ITS as the primary barcode marker. The length of the ITS region differs substantially among the different fungal genera and species, which distort the community description due to the preferential amplification of shorter sequences, thereby resulting in a biased quantification of the taxon relative abundances in the HTS research [39,40]. ITS1 and ITS2, which were shorter than the complete ITS, were of appropriate length for HTS and have become the most commonly used markers for fungal diversity analyses. Considering the lower phylogenetic richness and fewer OTUs generated by ITS1 region than those of the ITS2 region, ITS2 is recommended for metabarcoding [41]. In the present study, the results showed that the ITS2 region exhibited good identification ability for fungi in the ZSS samples. A total of 162 fungal taxa were detected in the ZSS samples. Among these fungal taxa, 92 cannot be resolved at the species level mainly due to the following reasons. First, some information on species is lacking on the UNITE database. Second, the ITS2 region cannot provide an adequate variation information in distinguishing some sibling species in *Aspergillus*, *Fusarium*, *Penicillium*, and *Candida* [42,43,44,45]. 

### 3.3. Prospects of Applying Amplicon Sequencing for the Analysis of Fungal Diversity in Herbal Materials

The application of HTS platform in analyzing the abundance and richness of the microbial species with low abundances in environmental samples overcomes the isolation limitations of the culture-based approach. Most microorganisms cannot be cultivated using traditional cultivation techniques [46]. Xia et al. [47] identified 55 fungal genera from Chinese Cordyceps through Illumina Miseq sequencing that were not observed using culture-dependent methods. The amplicon sequencing platform is widely used for the analysis of microbial diversity in soil, outdoor air, food, and other environmental samples [32,48,49,50,51]. The application of HTS platform in analyzing the fungal diversity in herbal materials has not been reported to date. In the present study, we utilized the amplicon sequencing platform and targeted the ITS2 sequences to investigate the fungal contamination in herbal materials for the first time. Fungal contamination in the ZSS samples is extremely common. Hence, all samples were infected. HTS platform is a good tool for investigating the occurrence of fungi, especially the potential toxigenic fungi, in herbal materials. The application of amplicon sequencing provides a new approach to analyze the fungal diversity in herbal materials, thereby providing an early warning for mycotoxin contamination in herbal materials.

## 4. Materials and Methods

### 4.1. Sampling

A total of 14 commercial ZSS samples were collected from Henan, Shanxi, Hebei, Liaoning, and Shandong, which are the five main ZSS producing provinces in China. The samples were divided into five groups, namely, SX, HN, LN, HB, and SD, by production areas. Among these commercial samples, six samples were affected with mildew due to inappropriate storage. Samples were also divided into two groups, namely, FM and WM, according to the presence or absence of macroscopic molds. The detailed information of the samples is listed in Table 1. 

### 4.2. DNA Extraction

Approximately 3 g ZSS samples were transferred into a 15 mL sterilized centrifuge tube with 10 mL of sterilized water and shaken with a vortex mixer for 20 min. Then, the mixture was filtered through a single layer of sterile gauze, and the microorganisms were collected from the filtrate by centrifugation at 7830 rpm for 15 min (Centrifuge 5430 R, Eppendorf AG, Hamburg, Germany) [52]. The total DNA was extracted using the EZNA^®^ soil DNA kit (Omega Bio-tek., Inc., Norcross, GA, USA) according to the manufacturer’s instructions.

### 4.3. PCR Amplification and HTS 

ITS2 sequences were amplified with the primer pairs ITS3 (5′-GCATCGATGAAGAACGCAGC-3′) and ITS4 (5′-TCCTCCGCTTATTGATATGC-3′) [53]. PCR was performed with the following conditions: Initial denaturation at 95 °C for 5 min; 35 cycles of denaturation at 95 °C for 45 s, annealing at 58 °C for 50 s, and elongation at 72 °C for 45 s; and a final extension at 72 °C for 10 min. Amplifications were conducted for each sample in triplicate, and the PCR products were pooled to minimize the PCR bias. The PCR products were detected using electrophoresis on a 2% agarose gel and purified using the DNA gel extraction kit (Axygen, Union City, CA, USA). Subsequently, the PCR products were further identified on a 2% agarose gel and quantified using QuantiFluorTM-ST (Promega, Madison, WI, USA). Then, amplicons were sequenced using the Illumina MiSeq PE250 platform (Illumina, San Diego, CA, USA) by AuwiGene Technology Co., Ltd. (Beijing, China). Raw sequences are available in the Sequence Read Archive of the NCBI under the accession numbers SAMN10275058–SAMN10275071.

### 4.4. Sequence Analysis

Raw FASTQ files were demultiplexed, and their quality was filtered using the Quantitative Insights into Microbial Ecology (QIIME, version 1.8, http://qiime.org) software [54]. The reads were truncated at any site, thereby receiving an average quality score of <20 over a 50 bp sliding window. The primers were exactly matched, thereby allowing two nucleotide mismatches, and the reads containing ambiguous bases were removed. Only sequences that overlap longer than 10 bp were merged according to their overlap sequences. The sequences were clustered into OTUs at 97% sequence similarity by using UPARSE (version 7.1, http://drive5.com/uparse/) [55], and chimeras were removed using USEARCH (version 8.1.1861, Http://Www.Drive5.Com/Usearch/) [56]. OTUs were denominated at the kingdom, phylum, class, order, family, genus, and species levels according to the UNITE database [57]. To estimate the α-diversity, we calculated the three metrics, including Chao 1, Shannon, and Good’s coverage, after the reads were normalized to the minimum reads (35,924 reads) in each sample. β-Diversity was measured using the weighted UniFrac and unweighted UniFrac distance matrices. PCoA was performed based on the weighted UniFrac distance matrices. Samples were also hierarchically clustered based on the unweighted Unifrac distances by using Unweighted Pair Group Method with Arithmetic Mean (UPGMA). LEfSe was applied to identify the differentially abundant taxa between the FM and WM groups [58]. Statistical differences among the sample groups were tested using ANOSIM (available through QIIME). Venn diagram and heat map were drawn using R tools.

## Figures and Tables

**Figure 1 toxins-10-00494-f001:**
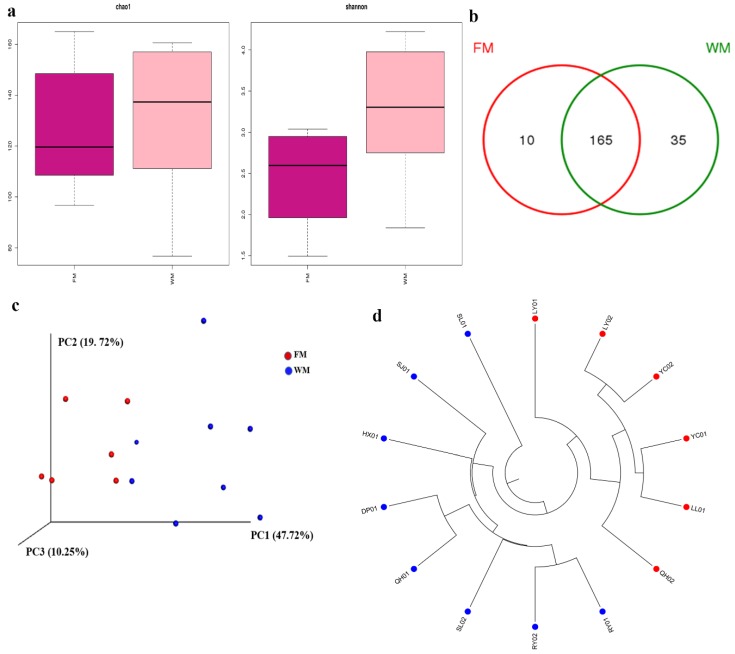
Diversity analyses of the fungal community in the Ziziphi Spinosae Semen (ZSS) samples. (**a**) α-Diversity plots of taxa richness (Chao 1 estimate) and diversity (Shannon index) in the FM and WM groups. (**b**) Venn diagram of operational taxonomic units (OTUs) in the FM and WM groups. (**c**) principal coordinate analysis (PCoA) plots on the basis on weighted UniFrac distance matrices. (**d**) Unweighted Pair Group Method with Arithmetic Mean (UPGMA) clustering on the basis of unweighted UniFrac distance analysis.

**Figure 2 toxins-10-00494-f002:**
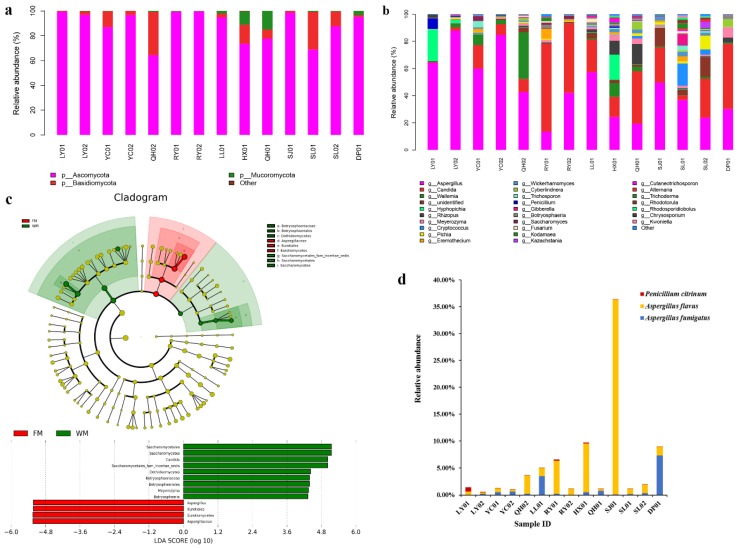
Composition analyses of the fungal communities in the ZSS samples. (**a**) Fungal composition at the phylum level. (**b**) Fungal composition at the genus level. (**c**) Fungal taxa that were differentially abundant in the FM and WM groups visualized using linear discriminant analysis effect size (LEfSe) analysis. (**d**) Relative abundances of the potential toxigenic fungi detected in the ZSS samples.

**Table 1 toxins-10-00494-t001:** Voucher information and GenBank accession numbers for the ZSS samples in this study.

Name	Voucher No.	Sources	Group	Mildewy	Group	GenBank Accession No.
Ziziphi Spinosae Semen	RY01	Ruyang, Henan	HN	No	WM	SAMN10275060
Ziziphi Spinosae Semen	RY02	Ruyang, Henan	HN	No	WM	SAMN10275061
Ziziphi Spinosae Semen	HX01	Hui, Henan	HN	No	WM	SAMN10275062
Ziziphi Spinosae Semen	SJ01	Shijiazhuang, Hebei	HB	No	WM	SAMN10275069
Ziziphi Spinosae Semen	QH01	Qinhuangdao, Hebei	HB	No	WM	SAMN10275067
Ziziphi Spinosae Semen	SL01	Shengli, Liaoning	LN	No	WM	SAMN10275065
Ziziphi Spinosae Semen	SL02	Shengli, Liaoning	LN	No	WM	SAMN10275066
Ziziphi Spinosae Semen	DP01	Dongping, Shandong	SD	No	WM	SAMN10275070
Ziziphi Spinosae Semen	LL01	Lanling, Shandong	SD	Yes	FM	SAMN10275071
Ziziphi Spinosae Semen	QH02	Qinhuangdao, Hebei	HB	Yes	FM	SAMN10275068
Ziziphi Spinosae Semen	YC01	Yuncheng, Shanxi	SX	Yes	FM	SAMN10275058
Ziziphi Spinosae Semen	YC02	Yuncheng, Shanxi	SX	Yes	FM	SAMN10275059
Ziziphi Spinosae Semen	LY01	Lingyuan, Liaoning	LN	Yes	FM	SAMN10275063
Ziziphi Spinosae Semen	LY02	Lingyuan, Liaoning	LN	Yes	FM	SAMN10275064

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
