# Peer review of "Analysis of the Fungal Community in Ziziphi Spinosae Semen through High-Throughput Sequencing"

_toxins, 2018, doi:10.3390/toxins10120494_

Reviewer 1 Report

The authors present an interesting study using the Illumina MiSeq PE250 platform and targeted the internal transcribed spacer 2 sequences to investigate fungal contamination in herbal materials (Ziziphi Spinosae Semen). The manuscript is well written, cites the relevant literature and presents the results clearly and illustrative. Just two minor remarks:

L 45: The sentence should be changed to “…because the ingestion of even extremely small amounts of mycotoxins …”

L 48: “AFs and OTA … are the most common occurring contaminants.” This general statement is doubtful. The occurrence of mycotoxins strongly depends on climate conditions and food matrix. For cereals such as wheat and barley, zearalenone and deoxynivalenol are prevalent mycotoxins. Also for herbs, other mycotoxins than AFs and OTA are common, e.g. fumonisins (Food Control 2019, 59: 63-70). Authors are requested to reconsider this sentence.

Author Response

Point 1: L 45: The sentence should be changed to “…because the ingestion of even extremely small amounts of mycotoxins …”

Response 1: The sentence has been revised as the reviewer suggested (page 2, line 47).

Point 2: L 48: “AFs and OTA … are the most common occurring contaminants.” This general statement is doubtful. The occurrence of mycotoxins strongly depends on climate conditions and food matrix. For cereals such as wheat and barley, zearalenone and deoxynivalenol are prevalent mycotoxins. Also for herbs, other mycotoxins than AFs and OTA are common, e.g. fumonisins (Food Control 2019, 59: 63-70). Authors are requested to reconsider this sentence.

Response 2: We are grateful for the reviewer’s comments. The sentence has been revised and the corresponding references have been updated (page 2, lines 48-50; page 8, lines 296-299).

Reviewer 2 Report

This manuscript is focused on the application of HTS platform in analyzing the abundance and richness of the fungal species on Ziziphi Spinosae Semen. Recently, microbiota analysis through amplicon sequencing approaches are increasingly gaining attention in microbiology. The use of ITS2 represents a good compromise for metabarcoding, even if in general the identification at species level is sometimes not possible.

Below are listed some suggestions:

Line 44..46: Concerning the ingestion of mycotoxins authors should include a short sentence and at least one reference about accumulation of mycotoxins with the diet and their deleterious cumulative effect.

Line 48..50: include also classification of OTA according to IARC

Line 51: replace “Some fungi..” with “Some species…”; Fusarium and Alternaria genera are not reported as AFs and OTA producers, indeed since the sentence is focused on AFs and OTA producers the reference to Fusarium and Alternaria genera

Line 55: replace “…Aspergillus flavus are AFB1-positive.”  with “…Aspergillus flavus were AFB1-positive.”

Line 56: replace “These medicinal plant samples predominantly consist of..” with “ The frequently isolated fungi predominantly consist of…”

Line 60: “…toxigenic fungi belonging to Aspergillus, Penicillium, ….”

Line 72..74: limits for AFs contamination should be indicated

Line 124: “were higher than….”

Line 157..160: “In contrast to previous studies, Penicillium species were  detected at relatively low abundances (<1%) in all ZSS samples, except for the LY01 sample (7.77%). This result may be attributed to the fact that different herbal medicines are susceptible to infection by different fungi types.” This difference in Penicillium detection could be related to the different matrix and storage conditions, probably inappropriate. Authors should extend the discussion.

Line 214..217: “Among these commercial samples, six samples were affected with mildew due to inappropriate storage. Samples were also divided into two groups, namely, FM and WM, according to the presence or absence of macroscopic molds.” The presence of macroscopic molds on samples of group FM exerts an aberrant effect on the capability of detection of a wide fungi diversity. This influence is reflected by diversity indices.

FigureS2: the captions should be rephrased eliminating any reference to significant comparisons.

Author Response

Point 1: Line 44..46: Concerning the ingestion of mycotoxins authors should include a short sentence and at least one reference about accumulation of mycotoxins with the diet and their deleterious cumulative effect.

Response 1: We appreciate the reviewer’s comment. A sentence and its corresponding references about accumulation of mycotoxins with the diet and their deleterious cumulative effect have been added (page 2, lines 44-45; page 8, lines 287-290).

Point 2: Line 48..50: include also classification of OTA according to IARC

Response 2: The classification of OTA according to IARC has been added (page 2, lines 52-53).

Point 3: Line 51: replace “Some fungi..” with “Some species…”; Fusarium and Alternaria genera are not reported as AFs and OTA producers, indeed since the sentence is focused on AFs and OTA producers the reference to Fusarium and Alternaria genera

Response 3: We are grateful for the reviewer’s comments. Accordingly, “Some fungi..” has been replaced with “Some species…”. “Fusarium and Alternaria” has been removed from the sentence (page 2, line 54).

Point 4: Line 55: replace “…Aspergillus flavus are AFB1-positive.”  with “…Aspergillus flavus were AFB1-positive.”

Response 4: The sentence has been revised as the reviewer suggested (page 2, line 57).

Point 5: Line 56: replace “These medicinal plant samples predominantly consist of..” with “ The frequently isolated fungi predominantly consist of…”

Response 5: The sentence has been revised as the reviewer suggested (page 2, line 59).

Point 6: Line 60: “…toxigenic fungi belonging to Aspergillus, Penicillium, ….”

Response 6: The sentence has been revised as the reviewer suggested (page 2, line 63).

Point 7: Line 72..74: limits for AFs contamination should be indicated

Response 7: The limits for AFs contamination in ZSS have been added (page 2, lines 75-77).

Point 8: Line 124: “were higher than….”

Response 8: The sentence has been revised as the reviewer suggested (page 4, line 128).

Point 9: Line 157..160: “In contrast to previous studies, Penicillium species were  detected at relatively low abundances (<1%) in all ZSS samples, except for the LY01 sample (7.77%). This result may be attributed to the fact that different herbal medicines are susceptible to infection by different fungi types.” This difference in Penicillium detection could be related to the different matrix and storage conditions, probably inappropriate. Authors should extend the discussion.

Response 9: We appreciate the reviewer’s comment. The discussion has been expanded as the reviewer suggested (page 5, lines 161-166).

Point 10: Line 214..217: “Among these commercial samples, six samples were affected with mildew due to inappropriate storage. Samples were also divided into two groups, namely, FM and WM, according to the presence or absence of macroscopic molds.” The presence of macroscopic molds on samples of group FM exerts an aberrant effect on the capability of detection of a wide fungi diversity. This influence is reflected by diversity indices.

Response 10: We are grateful for the reviewer’s comments. To assure the microorganisms are fully collected, approximately 3 g ZSS samples were transferred into a 15 ml sterilized centrifuge tube with 10 ml of sterilized water and shaken with a vortex mixer for 20 min. Then, the mixture was filtered through a single layer of sterile gauze, and the microorganisms were collected from the filtrate by centrifugation at 7,830 rpm for 15 min (Centrifuge 5430, Eppendorf AG, Germany). Therefore, the results obtained could reflect the level of fungal diversity in ZSS. As for the comparatively lower diversity of FM group than WM group, we speculate that the molds proliferate during storage and become the predominant fungi in moldy ZSS samples, thus inhibiting the growth of other fungi.

Point 11: Figure S2: the captions should be rephrased eliminating any reference to significant comparisons.

Response 11: The captions of Figure S2 have been rephrased as the reviewer suggested. 

Reviewer 3 Report

Dear author,

This is a nice piece of work. This study is extensive and performed very well. The conclusions are very useful to protect human health, especially while herbal medicines are increasingly used in the worldwide.

Author Response

We are grateful for the reviewer’s comments.